# Bipyridine Ruthenium(II) Complexes with Halogen-Substituted Salicylates: Synthesis, Crystal Structure, and Biological Activity

**DOI:** 10.3390/molecules28124609

**Published:** 2023-06-07

**Authors:** Martin Schoeller, Milan Piroš, Miroslava Litecká, Katarína Koňariková, Flóra Jozefíková, Alexandra Šagátová, Eva Zahradníková, Jindra Valentová, Ján Moncol

**Affiliations:** 1Department of Inorganic Chemistry, Slovak University of Technology, Radlinského 9, 812 37 Bratislava, Slovakia; martin.schoeller@stuba.sk (M.S.); milan.piros@stuba.sk (M.P.); flora.jozefikova@stuba.sk (F.J.); alexandra.sagatova@stuba.sk (A.Š.); 2Department of Materials Chemistry, Institute of Inorganic Chemistry of the CAS, Husinec-Řež č.p. 1001, 250 68 Řež, Czech Republic; litecka@iic.cas.cz; 3Institute of Medical Chemistry, Biochemistry and Clinical Biochemistry, Faculty of Medicine, Comenius University, Sasinkova 2, 811 08 Bratislava, Slovakia; katarina.konarikova@fmed.uniba.sk; 4Department of Chemical Theory of Drugs, Faculty of Pharmacy, Comenius University, Kalinčiakova 8, 832 32 Bratislava, Slovakia; valentova@fpharm.uniba.sk; 5Department of Inorganic Chemistry, Faculty of Science, Palacký University, 17. Listopadu 12, 771 46 Olomouc, Czech Republic; eva.zahradnikova01@upol.cz

**Keywords:** Ruthenium(II), medicinal inorganic chemistry, NSAID, crystal structure, antiproliferative activity, salicylate, anticancer drugs, BSA, DNA

## Abstract

Ruthenium complexes currently represent a perspective subject of investigation in terms of potential anticancer therapeutics. Eight novel octahedral ruthenium(II) complexes are the subject of this article. Complexes contain 2,2′-bipyridine molecules and salicylates as ligands, differing in position and type of halogen substituent. The structure of the complexes was determined via X-ray structural analysis and NMR spectroscopy. All complexes were characterized by spectral methods—FTIR, UV–Vis, ESI-MS. Complexes show sufficient stability in solutions. Therefore, their biological properties were studied. Binding ability to BSA, interaction with DNA, as well as in vitro antiproliferative effects against MCF-7 and U-118MG cell lines were investigated. Several complexes showed anticancer effects against these cell lines.

## 1. Introduction

Salicylic acid, salicylates, and all subsequent drugs from the non-steroidal anti-inflammatory drugs (NSAID) family, initially called Aspirin-like drugs, have been used for decades to hundreds of years in the therapy of inflammatory-related diseases. Relatively recently, it was recognized that this group of structurally diverse drugs shares the same molecular mechasnism. The prime mechanism of action is the inhibition of cyclooxygenase (COX) enzymes, also responsible for the conversion of arachidonic acid to prostanoids, which regulate inflammatory responses [1,2]. Currently, three structurally different cyclooxygenases have been described. The first is COX-1, which is primarily associated with physiological function, such as regulation of renal blood flow, circulation of gastric mucosa, proper platelet function, cytoprotective effect, and so on [3,4,5]. On the other hand, COX-2 is activated during inflammatory processes and is involved in the processes of tumor invasion and metastasis formation [5,6]. Recent findings suggest that this simplification, although often sufficient, is not entirely correct and that both enzymes are involved in physiological and pathological processes [7]. In 2002, the existence of COX-3 was also published [8]. In addition to the important role of NSAIDs in anti-inflammatory therapy, attention has recently been paid to these agents from a cancer-prevention and therapy perspective. It has been for a long time known that chronic inflammatory processes in many cases increase the risk of tumor development. Therefore, NSAIDs can offer the potential for prevention [9]. However, alongside these facts, it is observed that NSAIDs can also play an important role in cancer therapy itself. It was observed that NSAIDs increase the susceptibility of tumors to convulsive cytostatic therapy, promote apoptotic metabolic pathways in tumors, and reduce the ability of tumors to migrate. In addition, a cytoprotective effect as well as beneficial anti-inflammatory effects reducing the impact of conventional chemotherapy have been observed [10,11]. Therefore, currently, research focuses on the study of complexes with NSAIDs as potential agents in cancer treatments. Several complexes of transition metals (including ruthenium, copper, and others) with NSAIDs have been studied, and several biological properties have been observed with the perspective of being potential anticancer drugs [12,13,14].

Jaymani et al. studied the properties of four dimeric copper(II) complexes with salicylates, finding both antimicrobial activity and the ability to interact with DNA. Studied complexes can be described with the formulae [Cu_2_(bipy)_2_(5-Me-Sald)_2_(ClO_4_)_2_], [Cu_2_(biim)_2_(5-Me-Sald)_2_(ClO_4_)_2_], [Cu_2_(bipy)_2_(5-Br-SalH)_2_](ClO_4_)_2_, and [Cu_2_(bipy)_2_(5-Br-SalH)_2_](ClO_4_)_2_ (where bipy = 2,2′-bipyridine, biim = 2,2′-biimidazole, 5-Me-Sald = 5-bromosalicylaldehyde(1-) and 5-Br-SalH = 5-bromosalicylate(1-)). Complexes show high ability to interact with DNA—the highest binding constant has a complex with coordinated 5-bromosalicylate and bipyridine as ligands [15]. Copper(II) salicylate complexes that exhibit nuclease-like properties have also been published—specifically, complexes with the formulae [Cu(H_2_O)_2_(5-Cl-Sal)(Neo)], [Cu(*μ*-Sal)(Neo)]_2_, and [Cu_2_(*μ*-5-Cl-Sal)(5-Cl-SalH)_2_(Neo)_2_]·EtOH (where 5-Cl-Sal = 5-chlorosalicylate(2-), 5-Cl-SalH = 5-chlorosalicylate(1-), and Sal = salicylate(2-)). The first two complexes exhibit nuclease activity, indicating the interaction with DNA molecules [16].

Ruthenium(II) complexes with NSAIDs as ligands are also an outstanding object of investigation since currently, ruthenium complexes represent one of the most promising groups of studied complexes in terms of metal-based anticancer drugs [14,17,18]. This fact is demonstrated by several ruthenium complexes that have reached various stages of clinical trials, namely NAMI-A (ImH)[Ru(Im)(DMSO)Cl_4_] [19], KP1019 (IndH)[Ru(Ind)_2_Cl_4_], KP1339 Na[Ru(Ind)_2_Cl_4_] [20], RAPTA-C [Ru(*η^6^*-p-cymene)Cl_2_(pta)] [21], and TLD1433 [Ru(dmb)_2_(IP-3T)]Cl_2_ [22] (where Im/ImH = Imidazole/ImidazoleH(1+), Ind/IndH = Indazole/IndazoleH(1+), pta = 1,3,5-triaza-7-phosphaadamantane, DMSO = dimethyl sulfoxide, dmp = 4,4′-dimethyl-2,2′-bipyridine, and IP-3T = 2-(2,2′:5′,2″-tertiophene-5-yl)-1,3,7,8-tetraaza-1H-cyclopenta[I]phenantrene). Chan et al. prepared and studied three ruthenium complexes with salicylate(2-) as ligands with the formulae [Ru(phen)_2_(Sal)], [Ru(dmp)_2_(Sal)] and [Ru(bipy)_2_(Sal)] (where phen = 1,10-phenantroline). Within this group of complexes, the complex with phenanthroline ligands showed the highest antiproliferative activity, namely against the A549 cell line (lung cancer cell line). This complex was also subjected to more detailed investigation of the mechanism of action, and it was found that ROS accumulation in cells occurs due to inhibition of thioredoxine reductase, and consequently, MAPK signaling pathways are activated and AKT metabolic pathways are suppressed. DNA damage due to ROS was also detected, causing cell cycle arrest in the G0/G1 phase [23,24]. Promising results were also observed in cases of complexes containing ligands with covalently bound NSAIDs, such as diclofenac and indomethacin (via a carboxyl group that does not affect activity). Ruthenium(II) organometallic “piano-stool” complexes with the those ligands exhibited antiproliferative activity on *cis*-platin-resistant cell lines (A2780cisR), opening up the possibility of using these drugs in second-line anticancer therapy [25]. School at al. studied a set of five octahedral complexes of the general formula [Ru(bipy)_2_(X,Y-Sald)](BF_4_) (where X,Y-Sald = 5-bromosalicylaldehyde, 3,5-dibromosalicylaldehyde, 5-chlorosalicylaldehyde, 3,5-dichlorosalicylaldehyde, and 3-bromo-5-chlorosalicylaldehyde), considering the effects of position and type of substituent on salicylaldehyde ligand on antiproliferative activity of this complexes. A significant effect of the type of halogen-substitution of salicylaldehyde on the biological activity of the complexes was found. Experiments suggest that complexes containing dihalogen-substituted salicylaldehyde ligands exhibit higher activity compared to monohalogen-substituted ligands. Complexes with chloro-substituted ligands showed higher selectivity but in contrast lower half maximum inhibition concentration compared to bromo-substituted ligands. The mechanism of action of these complexes appears to be ROS generation and subsequent mitochondrial dysfunction [26].

The article deals with the synthesis and structural characterization of eight ruthenium complexes with bipyridine and salicylic acid derivatives (fluoro-, chloro-, bromo-, and iodo- in positions 4 or 5). The complexes were characterized by physicochemical methods (ESI-MS, ^1^H-NMR, FTIR, UV–Vis, and conductometry). The interactions of these complexes with biomacromolecules, such as albumin (BSA) and DNA (ct-DNA), were studied experimentally. The antiproliferative activity of the complexes was tested against cancer cell lines, specifically against breast cancer (MCF-7) and glioma (U-118MG) cell lines, using MTT assay.

## 2. Results and Discussion

### 2.1. Synthesis, Stability and Spectroscopical Study of Prepared Complexes

Complexes with the general formula [Ru(bipy)_2_(Sal)] were obtained in good yields via reaction of the precursor complex [Ru(bipy)_2_Cl_2_] with corresponding sodium salicylate prepared in situ. All complexes have been purified via column chromatography. Compounds of sufficient purity were obtained, which was confirmed by ^1^H-NMR, ESI-MS, FTIR, and conductivity measurements in DMSO. The study of spectral properties in solutions confirms the structure and stability of these complexes in DMSO solutions.

#### 2.1.1. Infrared Spectroscopy

Analysis of the IR spectra confirmed their agreement with the solved structures or with the proposed model from other indirect methods (specifically for complex **6**). In all spectra, a broad diffusion band can be observed in the range 3384 cm^−^^1^–3234 cm^−^^1^, which is characteristic for O-H functional groups bound by intermolecular hydrogen bonds. It was the ethanol and water molecules that were incorporated into the crystals that were confirmed by these bands. The complexes contain numerous C-H aryl groups, which were exhibited by several bands in the 3103 cm^−^^1^–3064 cm^−1^ region. The coordination of the 2,2′-bipyridine ligand is confirmed by the *ν*(C=N) band at 1571 cm^−1^–1609 cm^−1^. Infrared spectroscopy also provides information on the mode of coordination of the carboxyl group of salicylate ligands. In all spectra, bands of *ν_as_*(COO^−^) are observed in the range of 1578 cm^−1^–1529 cm^−1^, and those of *ν_s_*(COO^−^) in the range of 1400 cm^−1^–1417 cm^−1^. These bands are of mixed character as they are overlaid by bands of vibrations originating from the aromatic groups. The observation of multiple bands can also be explained by the presence of different types of hydrogen bonds in which the carboxyl group may be involved. Because of this observation, it is not possible to use *δ* values to assess the coordination of carboxyl groups with complete confidence. The values of the differences of the two bands are *δ*(for 1) = 140 cm^−^^1^, *δ*(for 2) = 127 cm^−^^1^, *δ*(for 3) = 124 cm^−^^1^, *δ*(for 4) = 129 cm^−^^1^, *δ*(for 5) = 135 cm^−^^1^, *δ*(for 6) = 155 cm^−^^1^, *δ*(for 7) = 145 cm^−^^1^, and *δ*(for 8) = 139 cm^−^^1^. These values are much greater than the ionic complexes, confirming monodentate mode of coordination. A further band that confirms the coordination of the salicylate-ligands is the valence vibration of phenolic group *ν*(C-O), whose values in the spectra of the complexes can be observed in the region from 1233 cm^−^^1^ to 1253 cm^−^^1^. A band of low intensity can be observed in the low-wavenumber region, which—when compared to values from the literature—could be attributed to the Ru-N valence vibrations [27]. The values of the individual wavenumbers of the most significant bands are summarized in Table 1. FTIR spectra of all complexes are shown in the Appendix A.

#### 2.1.2. ^1^H-NMR Spectra of Prepared Complexes

The ^1^H-NMR spectra of the complexes agree with the structure obtained from X-ray crystallography or with the predicted molecular structure (complex **6**) and are shown in the Appendix A. Due to the fact that in the case of Complex **6**, it was not possible to prepare a suitable crystal for X-ray analysis. The molecular structure was obtained using the 2D COSY NMR spectrum (Figure 1). Due to the coordination of the bipyridine ligand to the ruthenium(II) central atom, which leads to the asymmetric octahedral complex, we can observe peaks for every individual proton at different chemical shifts. As was published before [23], we assume that signals from protons that are closer to the carboxylic group of the salicylate ligand have higher chemical shifts. This observation confirms the octahedral geometry of all complexes in DMSO solution. It should also be noted at this point that the spectra were measured at least 24 h after the solutions were prepared. Based on 2D NMR the peak of H16 proton (with chemical shift 8.98 ppm) interact with H15, which is in overlapped multiplet with a chemical shift of 7.69 ppm and also with proton H14 (peak at 8.01 ppm). A very weak interaction is also observed with proton H13. Due to the symmetry of the complex, very similar interactions are observed for hydrogen H1 and protons H2, H3, and H4. The signal with a chemical shift of 8.55 ppm can be assigned to protons H5 and H12. Those protons interact with protons H7 and H6 (for H5) and H10 and H11 (for H12), respectively. There is no interaction with protons H8 and H9 (multiplet at 7.56 ppm), the assignment of which is based on interaction with other protons. The signals of the HA protons (6.32 ppm) are split by the HB protons (6.70 ppm), which still interact with the HC proton (multiplet at 7.69 ppm). Those peaks were assigned to the salicylate ligand.

#### 2.1.3. Electronic Spectra, ESI-MS and Study of Solvatochromism

Mass spectroscopy provides very important information about the structure of complexes in solution. This information is crucial before studying any biological properties of the complexes. In the ESI-MS spectra of Complexes **1** to **8**, the most intense molecular peaks are those corresponding to the [M + H]^+^ and [M + Na]^+^ compositions. The isotopic patterns of these peaks are in good agreement with the simulated spectra. In the case of several complexes, the presence of a peak corresponding to [Ru(bipy)_2_ + Cl]^+^ is observed, which may confirm a higher affinity of the central atoms for the nitrogen ligands than for the *O*-donor ligands. Peaks corresponding to different adducts were also observed with the solvents used such as MeOH and DMSO. All mass spectra were measured 3 days after the solutions were made (DMSO was used to dissolve the complexes). The mass spectra of all complexes are shown in Appendix A. Combining the information from ^1^H-NMR, ESI-MS and the conductivity of the complexes’ solutions provides strong evidence confirming the sufficient stability of the complexes in DMSO solutions.

During the study of prepared complexes, we noticed color changes in the solutions depending on the polarity of the solvent used. Figure 2A shows the color changes of Complex **1** depending on the solvents used. To investigate the solvatochromic properties, Complex **1** was dissolved in solvents of different polarity to have the same concentration of this complex in the solutions. Subsequently, the electron spectra were measured; they were very similar in character regarding the presence of identical bands in the spectra. The only change was observed on two bands in the visible region whose positions nearly linearly change with increasing polarity of the solvent used. With increasing relative polarity of the solvent [28], these bands shifted to lower wavelengths (hypsochromic effect) (Figure 2B,C).

The electronic spectra of Complexes **1**–**8** and the precursor complex [Ru(bipy)_2_Cl_2_] were measured in the form of a nujol suspension deposited on Whatman paper. The solution spectra of the complexes (concentration 2 × 10^−5^ M) were measured in an aqueous solution containing 0.5% DMSO. The maxima of the individual bands in the spectra are summarized in Table 2. The first part of the table summarizes the maxima from the solution spectra followed by those from the solid-state spectra, with the shoulders shown in parentheses. The two bands at lower wavelength values correspond mainly to transitions between MOs localized on the chelated 2,2′-bipyridine ligand (ILCT). According to the literature data, these transitions should be of the type π(N-L)→π*(N-L). The bands at higher wavelengths should correspond to transitions to the singlet excited state. Specifically, these are π(Ru)→π*(N-L) transitions between the central atom and the nitrogen ligand. Spectra measured in the solid state are more complicated and usually contain a larger number of bands. In the visible regions of the spectra, additional MLCT bands of the type π(Ru)→π*(N-L) could theoretically be observed, as could MLCT bands through the π(Ru)→π*(Sal) transition. It should also be possible to observe forbidden *d→d* transitions within the central atom [26,29,30]. All measured spectra are shown in Appendix A.

### 2.2. Crystal and Molecular Structures of Prepared Complexes

All complexes crystallize in the form of hydrates or adducts with water and ethanol. The molecular structures of complexes **1**·3H_2_O·EtOH, **2**·2.6H_2_O·2EtOH, **3**·6H_2_O, and **4**·3H_2_O are shown in Figure 3. Complex **1**·3H_2_O·EtOH crystallizes in a monoclinic system with space group P2_1_/c. Complexes **2**·2.6H_2_O·2EtOH and **3**·6H_2_O crystallize in a trigonal system with the R$\bar{3}$ space group. The central ruthenium(II) atom in all complexes is coordinated in the shape of a tetragonal bipyramid. In all complexes, one salicylate ligand and two bipyridine ligands coordinate bidentately to the central ruthenium(II) atom. The chromophore of all complexes is therefore the same {RuO_2_N_4_} and is formed by one oxygen atom of the carboxyl group and one phenolic oxygen atom of the salicylate ligand. The other coordination sites are occupied by the four donor nitrogen atoms of the two coordinated bipyridines. The bond lengths between the ruthenium central atoms and the two oxygen donor atoms are very similar and range from 2.0587 Å to 2.070 Å. Bond lengths between ruthenium atoms and nitrogen atoms in range from 2.020 Å to 2.082 Å. Selected bond lengths in complexes **1**·3H_2_O·EtOH, **2**·2.6H_2_O·2EtOH, **3**·6H_2_O and **4**·3H_2_O are summarized in Table 3. There was also an incorporation of solvent molecules into the crystal structure of all complexes. In the case of complex **1**·3H_2_O·EtOH, two water molecules were incorporated onto the asymmetric moiety, with one molecule forming a bridge between the two complexes within the asymmetric moiety via hydrogen bonds. In the case of complex **2**·2.6H_2_O·2EtOH, two ethanol molecules and one water molecule have been incorporated onto the asymmetric moiety, and they are bonded via hydrogen bonds. In the case of complex **3**·6H_2_O, the two water molecules on the asymmetric part are not bound by hydrogen bonds.

In the case of Complex **7**, it was possible to solve the structure of two pseudopolymorphs (**7A**·1.75H_2_O and **7B**·H_2_O·EtOH), which were obtained via the same procedure but crystallized from different volumes of mother liquor. The molecular structures of complexes **5**·1.55H_2_O, **7A**·1.75H_2_O, **7B**·H_2_O·EtOH, and **8**·4H_2_O are shown in Figure 4. Complex **5**·1.55H_2_O crystallizes in a monoclinic system with space group *P*2_1_/c, and both polymorphs—**7A**·1.75H_2_O and **7B**·H_2_O·EtOH—crystallize in a triclinic system with space group *P*$\bar{1}$. As in the case of the previous four complexes, one salicylate ligand and two bipyridines are coordinated to one central ruthenium(II) atom, and the chromophore is thus {RuO_2_N_4_}.The lengths of the selected bonds are comparable to complexes containing a coordinated salicylate substituted at position 4. The selected bond lengths are shown in Table 4. The bond lengths between the ruthenium central atoms and the two oxygen donor atoms are in range from 2.053 Å to 2.091 Å. Bond lengths between ruthenium central atoms and nitrogen atoms in range from 2.000 Å to 2.055 Å. The same chromophore {RuO_2_N_4_} and salicylate ligand binding are observed in the complex molecule in the crystal structure of the complex [Ru(bipy)_2_(Sal)]·4H_2_O [31].

### 2.3. Hirshfeld Surfaces Analysis

To investigate the intermolecular interactions in the crystal structures of the complexes, the Hirshfeld surfaces of the complexes were calculated and analyzed. Figure 5 shows the 3D Hirshfeld surface mapped over d_norm_ and shape index for complex **5**·1.55H_2_O. Other surfaces together with corresponding fingerprint plots are shown in Appendix A. The intensive red spots on the d_norm_-mapped surface indicate the close-contact interactions, mainly hydrogen bonds of varying strength. The shape indices of the 3D Hirshfeld surfaces show alternating red and blue triangular areas for complexes. This observation confirms the presence of π···π stacking, which is shown for individual complexes in Appendix A.

Hirshfeld 2D fingerprints were used to quantify the contribution of individual proximal contacts to the total surface area. In Figure 5C, the graph showing the individual contributions to the total surface area is shown. The close contacts with the largest proportion on the Hirshfeld surface are weak van der Waals H···H interactions. Very weak hydrogen bonds related to C···H/H···C close contacts cover from 23.1% to 25.9% of the surface. The stronger interactions are X···H/H···X hydrogen bonds (where X is a halogen atom), which make up 5.6–12.3% of the total surface. The proportions of O···H/H···O-type close contacts account for 9.6–23.6% of the surface, reflecting hydrogen bonds with the solvent molecules present. The proportions of C···C close contacts do not reflect the presence or absence of π···π stacking.

### 2.4. Interaction with Bovine Serum Albumin

Albumin is the most abundant protein in vertebrate blood plasma (accounting for more than 60% of the total protein content of blood plasma). Studying the interaction of potential drugs with proteins present in blood is essential to elucidating the pharmacokinetics and pharmacodynamics of these agents [32]. Bovine serum albumin (BSA) was chosen as a model protein target since human serum albumin (HSA) showed marked instability in citrate and the TRIS buffer used in our experiments. The use of HSA would lead to significant errors and to an overestimation of the interaction ability of the studied substances. Due to the presence of tryptophan residues in the protein structure, BSA exhibits intense fluorescence, with a maximum at 336 nm (with our instrument setup and measurement conditions). The wavelength of the excitation radiation was 280 nm. All prepared complexes quenched the fluorescence of BSA, as can be observed in Figure 6B. The largest fluorescence quenching was caused by complex **8**, which can also be observed in the changes in the fluorescence spectra in Figure 6A.

The measured data were analyzed using the dependence of the relative intensity *I/I_0_* on the concentration of the complex, [complex], according to the Stern–Volmer equation (Equation (1)). The dependencies for individual complexes along with the fit to the Stern-Volmer equation are shown in Appendix A. Thus, the dynamic quenching constants (*K_SV_*) and the values of the quenching rate constants (*k_q_*) were obtained, and they provide information about the type of fluorescence quenching mechanism. Their values for Complexes **1**–**8** are summarized in Table 5. From the dependence of the relative change in quenching with respect to complex concentration (Δ*I/I_0_*)/[complex] on the relative change in quenching Δ*I/I_0_*, the values of the binding rate constants (*K*) of the individual complexes with BSA and the number of binding sites for the complexes (*n*) on the BSA molecule were obtained using the Scatchard equation (Equation (2)). These values are also listed in Table 5.

The observed values of the fluorescence quenching rate constants (*k_q_*) of the complexes are in the interval of 1.07 × 10^13^ (for **4**)*–*1.99 × 10^13^ (for **7**) M*^−^*^1^s*^−^*^1^, which are significantly higher values than the threshold value of 2 × 10^10^ M*^−^*^1^s*^−^*^1^, indicating a static quenching mechanism and the existence of affinity of the complexes to BSA. The values of the binding constants K are in the range of 7.23 × 10^4^ (for **5**)*–*36.52 × 10^4^ (for **8**), while the values that would indicate a reversible interaction with BSA tend to be in the range 10^5^ to 10^6^; this interval was found for other complexes with NSAID ligands [33]. Complexes **1** and **5** have values of this constant slightly lower than the ideal interval, and thus, we assume that the complex interacts with BSA (with respect to the values of n), but this interaction is weak. On the other hand, the other complexes can be considered capable of reversible interaction. A clear dependence of the ability to interact with BSA on the structure of the complexes can be seen in the values of the binding constants (Figure 5C). The ability of the complexes to interact with BSA increases in the group with halogen-substituted salicylate ligands in order from fluorine to iodine. The highest values of K were found in complexes with iodine (8). The position of the substituent also plays a significant role in the interaction with BSA. The higher values of K were obtained in complexes with a substituent on the fifth carbon relative to the carboxyl group of the ligand. The numbers of binding sites for the complexes on BSA range from 0.86 (Complex **4**) to 1.28 (Complex **5**).

### 2.5. Interaction with DNA

DNA molecules represent one of the numerous molecular targets of action of several antibiotic, antiviral, and anticancer drugs. Therefore, the ability of the prepared complexes to interact with DNA extracted from calf thymus (ct-DNA) was studied. The following experimental techniques were selected: titration of the complex solution with ct-DNA solution monitored using UV–Vis spectroscopy and study of the ability to quench the fluorescence of the ethidium bromide–DNA complex.

The ability of the complexes to interact with DNA was investigated by measuring changes in the electronic spectra with increasing amounts of added ct-DNA solution. Changes associated with increasing concentration of DNA in solution were observed in the spectra of all complexes. The band– located at the lowest wavelengths were used to calculate the binding constants using the Wolfe–Shimer equation (Equation (3)). This band should originate from the electron transition within the ligands. The bands at higher wavelengths come from MLCT transitions, which are less affected by changes in the energies of the orbitals localized on the ligands. The results for Complexes **1** to **8** are summarized in Table 6. The values of the binding constants lie in the interval from 3.966 × 10^3^ M*^−^*^1^ (for complex **6**) to 6.332 × 10^5^ M*^−^*^1^ (for complex **2**). In all the spectra of the complexes, hyperchromism of the band under consideration is observed, suggesting that these complexes probably interact electrostatically or by groove binding with DNA molecules [34].

The values of the calculated binding constants are shown graphically in Figure 7. It can be concluded that no significant trend is observed in their values with respect to the structure of the complexes. The figure also shows the changes in the spectra of Complex **2**, which has the highest binding constant. The other spectra are shown in the Appendix A.

Ethidium bromide exhibits a significant ability to interact with the DNA molecules via the intercalation mechanism. The ethidium bromide–DNA (EB-DNA) complex exhibits intense fluorescence, with a maximum at 614 nm after excitation and a wavelength of 515 nm under our experimental conditions. The study provides information on the ability of the complexes to intercalate into the double-stranded DNA structure, thereby replacing EB in its structure, manifested by quenching of the fluorescence of the EB-DNA complex. All studied complexes quenched the fluorescence of the EB-DNA complex. Figure 8 shows the changes in the EB-DNA fluorescence spectra upon the addition of Complex **2**, which caused up to 94.7% fluorescence quenching (changes in the fluorescence spectra upon addition of the other complexes are shown in Appendix A). The dependencies of the relative fluorescence intensity of the EB-DNA complex after additions of Complexes **1**–**8** are shown as a function of the relative concentration of the complex with respect to the DNA concentration (*r*) in Figure 8A.

The lowest fluorescence quenching rate constant was calculated for Complex **1**. Specifically, this value is *k_q_* = 1.368 × 10^12^ M^−1^s^−1^. On the contrary, the highest values were observed for Complex **2**, namely *k_q_* = 4.654 × 10^12^ M^−1^s^−1^. The values of calculated constants for Complexes **1**–**8** are summarized in Table 7. Since all values are significantly higher than the threshold value of 2 × 10^10^ M^−1^s^−1^, they clearly point to a static quenching mechanism. Thus, all the complexes demonstrated that intercalation is one of the key factors influencing the interaction mechanism of the complexes with DNA. In Figure 8B, the values of the dynamic quenching constants (*K**_SV_***) for the individual complexes **1**–**8** are presented in graphical form. No clear trend is observed that demonstrates that the position and type of substituent on the salicylate ligand affects the intercalation ability.

### 2.6. In Vitro Study of Anticancer Activity

Cancer cells (8 × 10^3^ cells/200 μL) were treated with several concentrations (2*–*10 × 10*^−^*^6^ M) of Complexes **1***–***8** for 24, 48, and 72 h, and the cytotoxic effects of the complexes were evaluated via MTT assay. All measurements were repeated twice using three parallels for each concentration of complex. Two types of cancer cell lines were selected, namely breast cancer cells (MCF-7) and glioblastoma cells (U-118MG). The IC(50) values obtained are summarized in Table 8. In general, MCF-7 cells were more sensitive to the complexes. A decrease in IC(50) values with increasing exposure time was observed for Complexes **1**, **3**, and **6**. In the case of Complexes **5** and **8**, degradation of MCF-7 cancer cells was observed even at the lowest used concentrations (2 × 10*^−^*^6^ M), so it was not possible to estimate the IC(50) values.

In the case of the MCF-7 cell line, after the shortest incubation time (24 h), Complex **7** showed the highest antiproliferative activity. The IC(50) value was found to be 4.23 × 10*^−^*^6^ M, and Figure 9A shows the dependence of cell viability (as a percentage) on the concentration of this complex for both cell lines. For longer incubation times, Complex **1** was the most effective, with IC(50) values found to be 4.75 × 10*^−^*^6^ M (for 48 h) and 3.30 × 10*^−^*^6^ M (for 72 h). In the case of the glioblastoma cell line U-118MG, only Complexes **5**, **7**, and **8** showed antiproliferative effects. It should be noted that all of them contain halogen substituent at carbon number 5 of the coordinated salicylate ligand. Complex **5** showed the lowest IC(50) value after 24 h incubation, and Complex **8** showed the lowest IC(50) value after 48 h incubation. Figure 9B shows the viability of the cell lines as a dependence on Complex **5**’s concentration.

## 3. Materials and Methods

### 3.1. Synthesis and Chemicals

All reagents and solvents used for synthesis and measurements were purchased in reagent grade from Acros Organics, Alfa Aesar, and Centralchem. Lithium chloride was dried by heating to approximately 300 °C for about two hours. Other solvents and chemicals were used without further purification. In the study of the interaction of complexes with BSA and DNA, a citrate buffer was used, which was prepared by dissolving 15 nM of sodium citrate and 150 nM of sodium chloride in distilled water. The pH of the buffer was adjusted to 7 using sodium hydroxide or hydrochloric acid.

*Synthesis of [Ru(bipy)_2_Cl_2_]*, The synthesis of precursor complex was based on a modified procedure described in the literature [35]. To 10 cm^3^ of dimethylformamide in round bottom flask were added 2,2′-bipyridine (30 mmol, 2 equiv., 4.69 g), RuCl_3_·xH_2_O (where x ≅ 3) (15 mmol, 1 equiv., 3.92 g) and dried lithium chloride (75 mmol, equiv., 6.39 g). The reaction mixture was allowed to reflux for eight hours in an oil bath. Subsequently, after cooling to laboratory temperature, 100 cm^3^ of acetone was added. The solution thus prepared was allowed to crystallize in a freezer for 24 h. The crude soil product was filtered, washed with distilled water, a small amount of ethanol, and diethyl ether. After drying, 8.31 g (57%) of solid product was obtained. FTIR (ATR) ν/cm^−^^1^: 3504, 3465, 3097, 3067, 1599, 1456, 1441, 1416, 1306, 1261, 1016, 763, 723, 424. UV–Vis (nujol) *λ_max_*/nm: 363, 544, 583(sh), 641(sh).

*General procedure for the synthesis of complexes*, The synthesis scheme is shown in the figure below (Figure 10). The precursor complex [Ru(bipy)_2_Cl_2_] (1 mmol, 1 equiv., 0.484 g) was dissolved in 50 cm^3^ of mixed solvent (ethanol:water—2:1 *v*:*v*). The second solution was prepared by dissolving a correspondent derivate of salicylic acid (1 mmol, 1 equiv.) in the same solvent and was subsequently neutralized with an excess of sodium hydroxide (2.3 mmol, 2.3 equiv., 0.092 g). The solutions thus prepared were mixed and allowed to reflux for three days. Subsequently, the reaction mixture was evaporated to dry using a rotary evaporator, and the product was purified via the column chromatography “dry-start method” using silica as the stationary phase and a mixture of acetonitrile and water (9:1, *v*:*v*) as the mobile phase. Single crystals suitable for X-ray crystallography were prepared via slow evaporation of solvent (ethanol:water—1:1, *v*:*v*).

*[Ru(bipy)_2_(4-F-Sal)]* (**1**), Yield 0.514 g (83%). FTIR (ATR) *ν*/cm^−^^1^: 3660, 3234, 3068, 2966, 1609, 1556, 1416, 1253, 1015, 986, 759, 725, 652, 608, 410. ^1^H-NMR (300 MHz, *d_6_*-DMSO, 298 K) *δ*/ppm: 9.00 (ddd, *J* = 5.7; 1.6; 0.7 Hz, 1H, A, bipy); 8.95 (ddd, *J* = 5.6; 1.6; 0.7 Hz, 1H, B, bipy); 8.69 (m, 2H, C, bipy); 8.57 (m, 2H, D, bipy); 8.03 (m, 2H, E, bipy); 7.71 (m, 5H, F, 4H bipy + 1H 4-F-Sal); 7.56 (m, 2H, G, bipy); 7.15 (m, 2H, H, bipy); 6.00 (m, 1H, I, 4-F-Sal); 5.92 (m, 1H, J, 4-F-Sal). ESI-MS (positive mode) *m*/*z*: {[Ru(bipy)_2_(4-F-Sal)]+H^+^} 569.064. Conductivity *Λ_M_*/S.cm^2^mol^−^^1^: 7.5 (in 1 mM solution of DMSO). UV–Vis (nujol) *λ_max_*/nm: 369, 396, 575, 688(sh).

*[Ru(bipy)_2_(4-Cl-Sal)]* (**2**), Yield 0.573 g (83%). FTIR (ATR) *ν*/cm^−^^1^: 3356, 3101, 3068, 2966, 2872, 1583, 1542, 1415, 1237, 923, 760, 726, 655, 600, 421. ^1^H-NMR (300 MHz, *d_6_*-DMSO, 298 K) *δ*/ppm: 8.99 (dd, *J* = 5.7; 1.5 Hz, 1H, A, bipy); 8.94 (dd, *J* = 5.8; 1.5 Hz, 1H, B, bipy); 8.68 (m, 2H, C, bipy); 8.55 (m, 2H, D, bipy); 8.01 (m, 2H, E, bipy); 7.69 (m, 5H, F, 4H bipy+1H 4-Cl-Sal); 7.56 (m, 2H, G, bipy); 7.14 (m, 2H, H, bipy); 6.30 (d, *J* = 2.2 Hz, 1H, I, 4-Cl-Sal); 6.13 (dd, *J* = 8.6; 2.2 Hz, 1H, J, 4-Cl-Sal). ESI-MS (positive mode) *m*/*z*: {[Ru(bipy)_2_(4-Cl-Sal)]+H^+^} 585.077. Conductivity *Λ_M_*/S.cm^2^mol^−^^1^: 6.4 (in 1 mM solution of DMSO). UV–Vis (nujol) *λ_max_*/nm: 368, 394, 571, 604, 708(sh).

*[Ru(bipy)_2_(4-Br-Sal)]* (**3**), Yield 0.502 g (77%). FTIR (ATR) *ν*/cm^−^^1^: 3354, 3101, 3068, 2966, 2872, 1578, 1540, 1416, 1234, 903, 760, 727, 420. ^1^H-NMR (300 MHz, *d_6_*-DMSO, 298 K) *δ*/ppm: 8.97 (m, 1H, A, bipy); 8.93 (d, *J* = 5.6 Hz, 1H, B, bipy); 8.70 (m, 2H, C, bipy); 8.58 (m, 2H, D, bipy); 8.04 (m, 2H, E, bipy); 7.70 (m, 4H, F, bipy); 7.62 (d, *J* = 8.5 Hz, 1H, G, 4-Br-Sal); 7.56 (m, 2H, H, bipy); 7.16 (m, 2H, I, bipy); 6.44 (d, *J* = 2.1 Hz, 1H, J, 4-Br-Sal); 6.24 (dd, *J* = 8.4; 2.1 Hz, 1H, K, 4-Br-Sal). ESI-MS (positive mode) *m*/*z*: {[Ru(bipy)_2_(4-Br-Sal)]+H^+^} 630.910. Conductivity *Λ_M_*/S.cm^2^mol^−^^1^: 7.5 (in 1 mM solution of DMSO). UV–Vis (nujol) *λ_max_*/nm: 364, 393, 575, 716(sh).

*[Ru(bipy)_2_(4-I-Sal)]* (**4**), Yield 0.598 g (84%). FTIR (ATR) *ν*/cm^−^^1^: 3294, 3064, 1571, 1542, 1413, 1245, 1145, 1015, 888, 759, 726, 656, 597, 421. ^1^H-NMR (300 MHz, *d_6_*-DMSO, 298 K) *δ*/ppm: 8.98 (ddd, *J* = 5.6; 1.6; 0.7 Hz, 1H, A, bipy); 8.93 (ddd, *J* = 5.6; 1.6; 0.8 Hz, 1H, B, bipy); 8.70 (m, 2H, C, bipy); 8.57 (m, 2H, D, bipy); 8.04 (m, 2H, E, bipy); 7.71 (m, 4H, F, bipy); 7.57 (m, 2H, G, bipy); 7.43 (d, *J* = 8.3 Hz, 1H, H, 4-I-Sal); 7.15 (m, 2H, I, bipy); 6.68 (d, *J* = 1.9 Hz, 1H, J, 4-I-Sal); 6.46 (dd, *J* = 8.4; 1.9 Hz, 1H, K, 4-I-Sal). ESI-MS (positive mode) *m*/*z*: {[Ru(bipy)_2_(4-I-Sal)]+H^+^} 676,905. Conductivity *Λ_M_*/S.cm^2^mol^−^^1^: 13.5 (in 1 mM solution of DMSO). UV–Vis (nujol) *λ_max_*/nm: 360, 393, 573, 711(sh).

*[Ru(bipy)_2_(5-F-Sal)]* (**5**), Yield 0.542 g (81%). FTIR (ATR) *ν*/cm^−^^1^: 3383, 3072, 1566, 1547, 1459, 1412, 1321, 1233, 1412, 1321, 1233, 1121, 1015, 784, 758, 726, 654, 424. ^1^H-NMR (300 MHz, *d_6_*-DMSO, 298 K) *δ*/ppm: 8.97 (m, 2H, A, bipy); 8.67 (m, 2H, B, bipy); 8.55 (m, 2H, C, bipy); 8.00 (m, 2H, D, bipy); 7.68 (m, 4H, E, bipy); 7.56 (m, 2H, F, bipy); 7.35 (dd, *J* = 11.2; 3.6 Hz, 1H, G, 5-F-Sal); 7.13 (m, 2H, H, bipy); 7.58 (m, 1H, I, 5-F-Sal); 6.27 (dd, *J* = 9.1; 5.0 Hz, 1H, J, 5-F-Sal). ESI-MS (positive mode) *m*/*z*: {[Ru(bipy)_2_(5-F-Sal)]+H^+^} 569.182. Conductivity *Λ_M_*/S.cm^2^mol^−^^1^: 9.8 (in 1 mM solution of DMSO). UV–Vis (nujol) *λ_max_*/nm: 395, 573, 725(sh).

*[Ru(bipy)_2_(5-Cl-Sal)]* (**6**), Yield 0.465 g (80%). FTIR (ATR) *ν*/cm^−^^1^: 3384, 3066, 1589, 1557, 1456, 1402, 1311, 1241, 1014, 825, 758, 720, 656, 420. ^1^H-NMR (300 MHz, *d_6_*-DMSO, 298 K) *δ*/ppm: 8.98 (m, 1H, A, bipy); 8.93 (m, 2H, B, bipy); 8.68 (m, 2H, C, bipy); 8.55 (m, 2H, D, bipy); 8.01 (m, 2H, E, bipy); 7.69 (m, 5H, F, 4H bipy + 1H 5-Cl-Sal); 7.56 (m, 2H, G, bipy); 7.14 (m, 2H, H, bipy); 6.70 (dd, *J* = 8.9; 3.1 Hz, 1H, I, 5-Cl-Sal); 6.32 (d, *J* = 8.9 Hz, 1H, J, 5-Cl-Sal). ESI-MS (positive mode) *m*/*z*: {[Ru(bipy)_2_(5-Cl-Sal)]+H^+^} 584.973 Conductivity *Λ_M_*/S.cm^2^mol^−^^1^: 6.6 (in 1 mM solution of DMSO). UV–Vis (nujol) *λ_max_*/nm: 389, 503, 602, 719(sh).

*[Ru(bipy)_2_(5-Br-Sal)]* (**7**), Yield 0.459 g (66%). FTIR (ATR) *ν*/cm^−^^1^: 3320, 3064, 2964, 1584, 1545, 1457, 1440, 1400, 1318, 1243, 1139, 1014, 822, 758, 655, 421. ^1^H-NMR (300 MHz, *d_6_*-DMSO, 298 K) *δ*/ppm: 8.98 (ddd, *J* = 5.6; 1.5; 0.7 Hz, 1H, A, bipy); 8.92 (ddd, *J* = 5.6; 1.6; 0.7 Hz, 1H, B, bipy); 8.68 (m, 2H, C, bipy); 8.55 (m, 2H, D, bipy); 8.01 (m, 2H, E, bipy); 7.79 (d, *J* = 3.0 Hz, 1H, F, 5-Br-Sal); 7.69 (m, 4H, G, bipy); 7.56 (m, 2H, H, bipy); 7.13 (m, 2H, I, bipy); 6.80 (dd, *J* = 8.9; 3.0 Hz, 1H, J, 5-Br-Sal); 6.28 (d, *J* = 8.9 Hz, 1H, K, 5-Br-Sal). ESI-MS (positive mode) *m*/*z*: {[Ru(bipy)_2_(5-Br-Sal)]+H^+^} 630,920. Conductivity *Λ_M_*/S.cm^2^mol^−^^1^: 8.4 (in 1 mM solution of DMSO). UV–Vis (nujol) *λ_max_*/nm: 601.

*[Ru(bipy)_2_(5-I-Sal)]* (**8**), Yield 0.587 g (78%). FTIR (ATR) *ν*/cm^−^^1^: 3359, 3067, 1578, 1556, 1456, 1396, 1311, 1250, 1014, 825, 757, 724, 528, 421. ^1^H-NMR (300 MHz, *d_6_*-DMSO, 298 K) *δ*/ppm: 8.97 (ddd, *J* = 5.6; 1.5; 0.7 Hz, 1H, A, bipy); 8.91 (ddd, J = 5.6; 1.5; 0.7 Hz, 1H, B, bipy); 8.67 (m, 2H, C, bipy); 8.55 (m, 2H, D, bipy); 8.00 (m, 3H, E, 2H bipy+1H 5-I-Sal); 7.69 (m, 4H, F, bipy); 7.56 (m, 2H, G, bipy); 7.13 (m, 2H, H, 5-I-Sal); 6.92 (dd, *J* = 8.8; 2.7 1H, I, 5-I-Sal); 6.19 (d, *J* = 8.8 Hz, 1H, J, 5-I-Sal). ESI-MS (positive mode) *m*/*z*: {[Ru(bipy)_2_(5-I-Sal)]+H^+^} 676.935. Conductivity *Λ_M_*/S.cm^2^mol^−^^1^: 14.9 (in 1 mM solution of DMSO). UV–Vis (nujol) *λ_max_*/nm: 391, 496, 595, 742(sh).

### 3.2. X-ray Crystallography

The data collection and cell refinement of Complex **5**·1.55H_2_O·EtOH were carried out using a Stoe StadiVari diffractometer with a Pilatus3R 300K HPD detector. Xenocs Genix3D Cu HF (microfocused sealed tube, λ = 1.54186 Å) was used as an X-ray source. The multi-scan absorption corrections were applied using the program Stoe LANA [36]. The data collection of Compounds **1**·3H_2_O·EtOH, **2**·2.6H_2_O·2EtOH, **3**·6H_2_O, **4**·3H_2_O, **7A**·1.75H_2_O, **7B**·H_2_O·EtOH, and **8**·4H_2_O was carried out using a Rigaku XtalLAB Synergy Dualflex diffractometer equipped with an HPD HyPix detector. PhotonJet Cu X-ray source has been used. Additionally, multi-scan absorption corrections were applied using CrysAlisPro software [37]. The diffraction intensities were corrected for Lorentz and polarization factors. The structure was solved using ShelXT [38] program and refined using the full-matrix least squares procedure with ShelXL (version 2018/3) [39]. Geometrical analyses were performed with ShelXL/Olex2.refine. The structures were drawn with OLEX2 [40]. The crystal data, conditions of data collection, and refinement are reported in Table 9.

### 3.3. Hirshfeld Surface Analysis

The software CrystalExplorer (ver. 21.5) [41] was used to calculate Hirshfeld surfaces [42] and associated fingerprint plots [43,44].

### 3.4. Molecular Spectroscopy

Infrared spectra were collected using NICOLET 5700 FTIR (Nicolet, Waltham, MA, USA) spectrometer using ATR technique at room temperature. UV–Vis spectra were measured on a SPECORD 250 Plus (Carl Zeiss Jena, Jena, Germany). ^1^H-NMR spectrum was collected with a Varian Unity-Inova (300 MHz). Chemical shifts are reported in ppm relative to DMSO as internal standard. For measurement of ESI-MS, an LCQ Fleet mass spectrometer (Thermo Scientific, Waltham, MA, USA) equipped with an electrospray ion source and a three-dimensional (3D) ion trap detector in the positive mode was used.

### 3.5. Interaction with Bovine Serum Albumin (BSA)

The ability of all the prepared complexes to interact with BSA was studied by quenching the BSA florescence. Citrate buffer was used as solvent, and the initial concentration of BSA was 30 μM. The quenching of the fluorescence of tryptophan residues at 336 nm was observed through gradual addition of the complexes’ solutions (10*^−^*^4^ M) to DMSO. The fluorescence emission spectra were recorded in the range of 300 nm to 420 nm. The wavelength of excitation radiation was 280 nm. The measured data were evaluated using the Stern–Volmer equation (Equation (1)), obtaining the values of the Stern–Volmer constant *K_SV_* (in M*^−^*^1^) and the BSA of the quenching constant *k_q_* (M*^−^*^1^s*^−^*^1^). Using the Scatchard equation (Equation (2)), the values of the binding constants *K_BSA_* (in M*^−^*^1^) and the values of the binding sites on albumin n were obtained.
(1)$$\frac{I_{0}}{I}=1+K_{SV}\left[Q\right]=1+k_{q}\tau_{0}\left[Q\right]$$
(2)$$\frac{\Delta I/I_{0}}{\left[Q\right]}=nK_{BSA}-K_{BSA}\frac{\Delta I}{I_{0}}$$
where *I* is the fluorescence intensity, *I*_0_ is the initial (before solutions of complexes addition) fluorescence intensity of BSA, *K_SV_* is the Stern–Volmer constant, [*Q*] (in M) is the concentration of the quencher, *k_q_* is the quenching constant (in M*^−^*^1^s*^−^*^1^), and *τ*_0_ is the lifetime of the emissive excitation state. *K_BSA_* (in M*^−^*^1^) is the bovine-serum-albumin binding constant, and *n* is the value of binging sides per albumin [45].

### 3.6. Interaction with ct-DNA

The ct-DNA was checked for sufficient protein purification before use. Using UV–Vis spectroscopy, the absorbances at two wavelengths (260 nm and 280 nm) were compared. The ratio of these two absorbances is less than 1.89, indicating that the ct-DNA is sufficiently protein-free [46]. The stock solution of DNA was prepared by dissolving 6 mg of ct-DNA in 5 cm^3^ of citrate buffer. Subsequently, the concentration of DNA in solution was determined using UV–Vis spectroscopy with a molar absorption coefficient of DNA at 260 nm (6600 M*^−^*^1^cm*^−^*^1^).

#### 3.6.1. Absorption Titrations

The interaction of complexes with DNA was studied using UV–Vis monitored titrations of complexes solutions in DMSO and buffer with DNA solution. The concentration of DMSO was kept at less than 1% (due to the instability of DNA in solutions with higher DMSO concentrations). The obtained data were evaluated using the Wolfe–Shimer equation (Equation (3)).
(3)$$\frac{\left[DNA\right]}{\varepsilon_{a}-\varepsilon_{f}}=\frac{\left[DNA\right]}{\varepsilon_{b}-\varepsilon_{f}}+\frac{1}{K_{b}(\varepsilon_{b}-\varepsilon_{f})}$$
where *K_b_* is the binging constant, [*DNA*] is the concentration of DNA in solution, *ε_f_* is the extinction coefficient of the free complex, *ε_f_* is the extinction coefficient of the fully bound form, and *ε_a_* is defined by *ε_a_* = A_obs_/[complex] [47,48].

#### 3.6.2. Quenching of the Fluorescence of EB-DNA Adduct

Another way of studying the interaction of complexes with DNA is to study the ability of complexes to displace EB from the EB-DNA complex. The EB-DNA adduct was prepared by mixing 20 μM of EB in buffer solution with buffer solution of 54 μM of DNA. The measurement was performed by sequential addition of DMSO solution of the complex to the EB-DNA adduct solution, while changes in the fluorescence spectra were monitored. The excitation wavelength was set to 515 nm, and spectra were recorded in the range of 550 nm to 800 nm. The Stern–Volmer equation (Equation (1)) was used to evaluate the measured data, similarly to the BSA interaction study [49].

### 3.7. Study of Anticancer Activity

#### 3.7.1. Cell Culture

Human breast cancer cells (MCF-7) and human glioblastoma cells (U-118MG) were purchased from the American Type Culture Collection (Manassas, VA, USA) and maintained in Dulbecco’s Modified Eagle Medium (DMEM, Life Technologies, Inc., Rockville, MD, USA) containing 10% fetal bovine serum, 100 μg/mL streptomycin, and 100 U/mL penicillin G at 37 °C in a humidified atmosphere of 5% CO_2_/95% air.

For experiments, cells were seeded on culture dishes or plates in amounts described below. Cells at passage numbers 10–13 were used.

#### 3.7.2. Cytotoxic Analysis

We have determined the cytotoxic effects of eight complexes (**1**–**8**) on carcinoma cells by using the MTT [3-(4,5-dimethylthiazol-2-yl)-2,5-diphenyltetrazolium bromide] colorimetric technique [50]. Cells were seeded (8 × 10^3^ cells/200 μL well) in individual wells of 96-multiwell plates. We added different concentrations of copper complexes (2–10 × 10^−4^ M) to the cells and incubated them for 24, 48, and 72 h at 37 °C (humidified atmosphere of 5% CO_2_/95% air). After 72 h, cells were treated with the MTT solution (5 mg/mL) in PBS (phosphate-buffered saline) (20 μL) for 4 h. The dark crystals of formazan formed in intact cells were dissolved in DMSO (dimethyl sulfoxide) (200 μL). The plates were shaken for 15 min and the optical density was determined at 490 nm using a MicroPlate Reader (Biotek, Winooski, VT, USA). All dye exclusion tests were performed three times.

## 4. Conclusions

The eight new complexes were prepared via reaction of the [Ru(bipy)_2_Cl_2_] complex with deprotonated salicylic acid derivatives. The yields of the syntheses ranged from 66% to 84%. The structures of seven complexes were solved by X-ray structural analysis. In the case of Complex **7**, a second pseudopolymorph was prepared and structurally characterized. All complexes have an octahedral shape of the coordination polyhedron. Two molecules of 2,2′-bipyridine and one salicylate(2-) ligand coordinate to the one ruthenium(II) central atom via one phenolic and one carboxylate oxygen atom. This leads to the formation of neutral complexes. π···π stacking interactions and hydrogen bonds are dominant in the supramolecular structure of prepared complexes. The purity of all complexes was confirmed, and the spectral properties of the complexes have also been studied. All the complexes showed solvatochromism, while in the case of Complex **1**, this phenomenon has been more closely studied. According to the analyses performed (MS and NMR), the complexes are sufficiently stable in DMSO and H_2_O solutions. In the solutions, the octahedral shape of the polyhedron was confirmed, and the molecular structure is preserved as in the solid state. Based on these observations, it was possible to study the biological properties of complexes. First, it was confirmed that the complexes can interact with BSA. Complex **8** shows the highest value of binding constant. The magnitude of the constant increases with increasing substituent (in order from fluorine to iodine) and substitution at carbon number 5 apparently increases the ability of the complexes to bind to BSA. Such dependence of the values of the binding constants on small changes in the structures of the complexes can be explained by the complex interacting with BSA via the salicylate ligand. The complexes appeared to be able to interact with DNA. Based on the ability of the complexes to capture the fluorescence of the EB-DNA complex, the intercalation mechanism is one of the main ones. The complexes also exhibit in vitro anticancer activity. More pronounced effects were found against the breast cancer cell line (MCF-7), with Complexes **5**, **7**, and **8** having IC(50) values less than 2 × 10^−6^ M. In addition, Complexes **5**, **7**, and **8** also have antiproliferative effects against the glioblastoma cell line U-118MG. In the future, we plan to study the electrochemical behavior of the complexes, considering that the ligand 2,2′-bipyridine represents a non-innocent ligand. We will also focus on a deeper study of the biological properties of the prepared complexes.

## Figures and Tables

**Figure 1 molecules-28-04609-f001:**
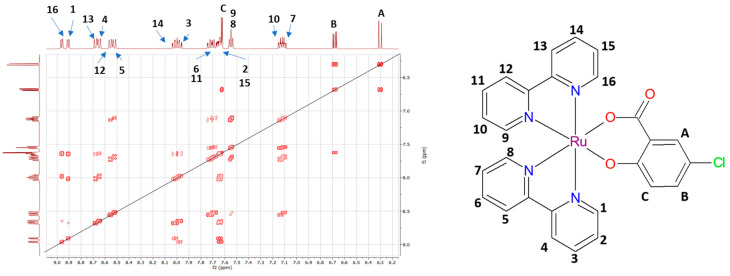
Two-dimensional COSY NMR spectrum of Complex **6** with the assignment of signals to molecular structures of the complex.

**Figure 2 molecules-28-04609-f002:**
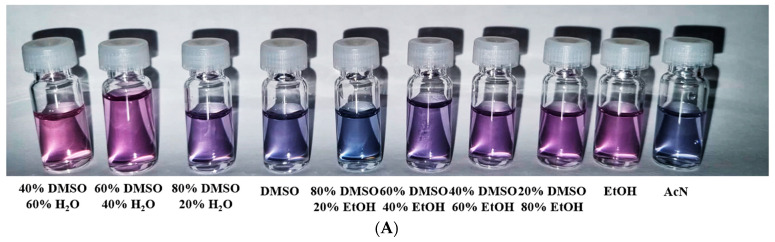
(**A**) The color of the solution of Complex **1** varies from blue to pink depending on the solvent used, (**B**) the measured electron spectra in different solvents, (**C**) the dependence of the wavelength of the band maxima in the UV–Vis spectra on the relative polarity of the solvent.

**Figure 3 molecules-28-04609-f003:**
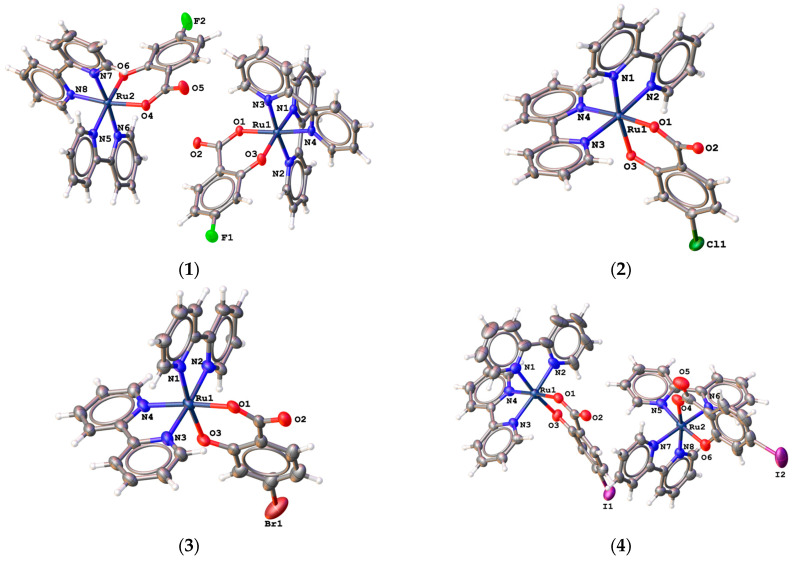
Molecular structure of complex molecules in crystal structures of [Ru(bipy)_2_(4-F-Sal)]·3H_2_O·EtOH (**1**·3H_2_O·EtOH), [Ru(bipy)_2_(4-Cl-Sal)]·2.6H_2_O·2EtOH (**2**·2.6H_2_O·2EtOH), [Ru(bipy)_2_(4-Br-Sal)]·6H_2_O (**3**·6H_2_O), and [Ru(bipy)_2_(4-I-Sal)]·3H_2_O (**4**·3H_2_O).

**Figure 4 molecules-28-04609-f004:**
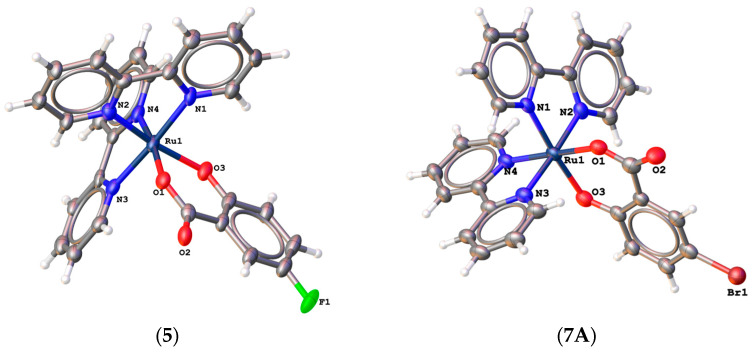
Molecular structure of complex molecules in crystal structures of [Ru(bipy)_2_(5-F-Sal)]·1.55H_2_O (**5**·1.55H_2_O), [Ru(bipy)_2_(5-Br-Sal)]·1.75H_2_O (**7A**·1.75H_2_O), [Ru(bipy)_2_(5-Br-Sal)]·H_2_O·EtOH (**7B**·H_2_O·EtOH), and [Ru(bipy)_2_(5-I-Sal)]·4H_2_O (**8**·4H_2_O).

**Figure 5 molecules-28-04609-f005:**
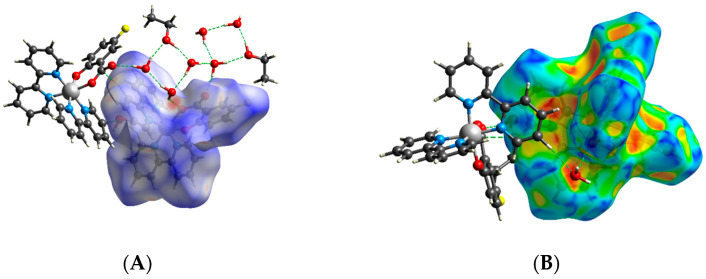
(**A**) d_norm_-mapped Hirshfeld surface for complex **5**·1.55H_2_O, (**B**) shape-index-mapped surface for **5**·1.55H_2_O, and (**C**) percentages of individual types of close contacts on the surface.

**Figure 6 molecules-28-04609-f006:**
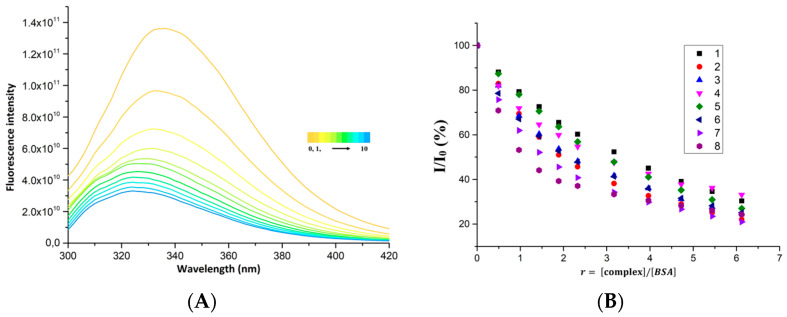
(**A**) Changes in fluorescence spectra of BSA upon Complex **8** concentration rising, (**B**) graphical dependence of relative BSA fluorescence emission intensity (*I*/*I*_0_) vs. concentration ratio $r=\left[\mathrm{complex}\right]/\left[BSA\right]$ for **1**–**8**, (**C**) graphical comparison of dynamic quenching constants (*K_SV_*) and binding constants (*K*) for **1**–**8**.

**Figure 7 molecules-28-04609-f007:**
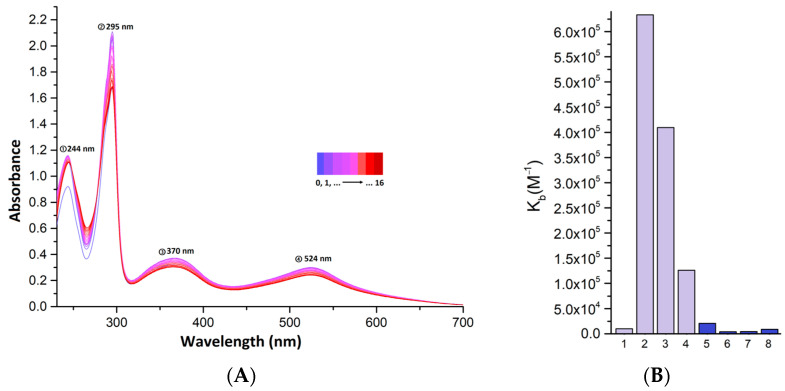
(**A**) changes in the electron spectra of Complex **2** upon addition of ct-DNA solution, (**B**) graphical comparison of binding constants for Complexes **1**–**8**.

**Figure 8 molecules-28-04609-f008:**
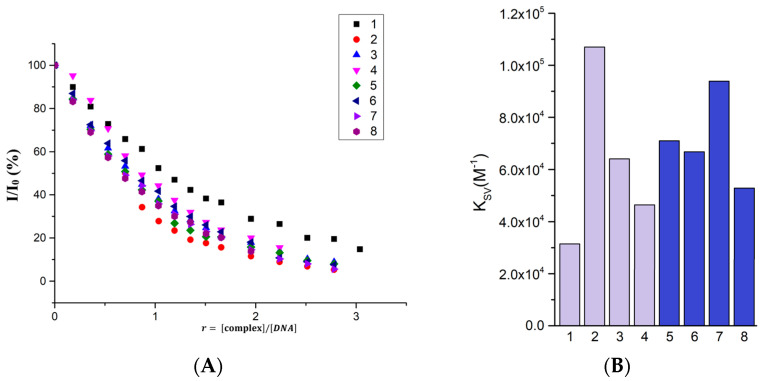
(**A**) Graphical dependence of relative EB-DNA fluorescence emission intensity (*I*/*I_0_*) vs. concentration ratio $r=\left[\mathrm{complex}\right]/\left[DNA\right]$ for **1**–**8**, (**B**) graphical comparation of dynamic quenching constants (*K_SV_*) for **1**–**8**, (**C**) changes in fluorescence spectra of EB-DNA complex upon Complex **2** concentration rising.

**Figure 9 molecules-28-04609-f009:**
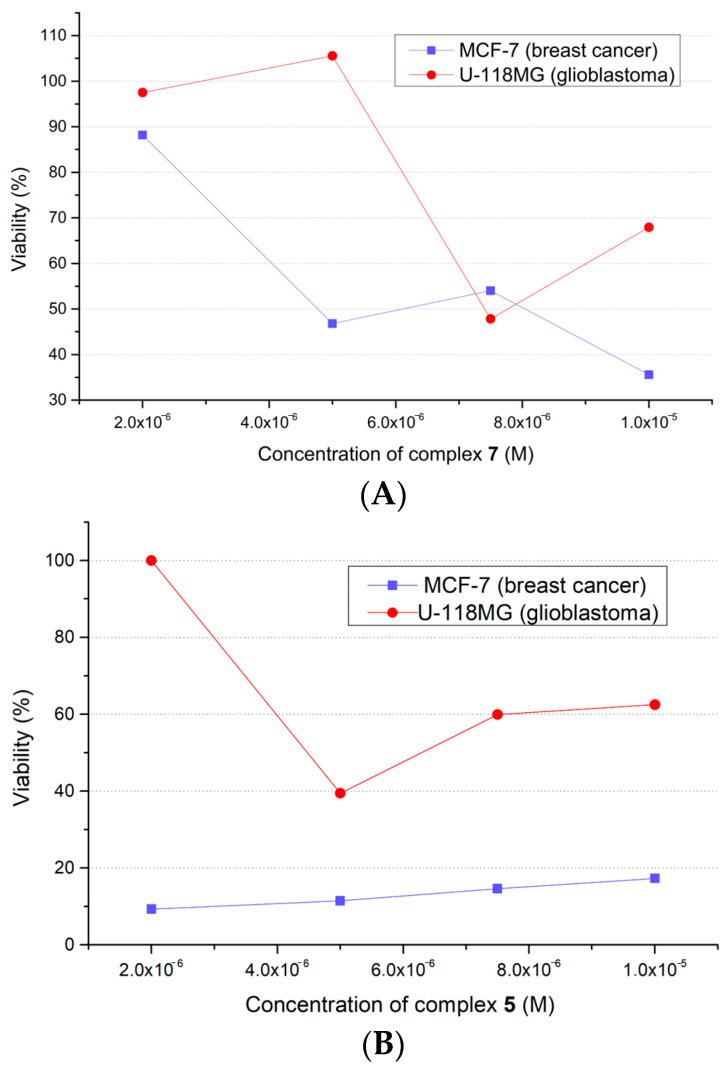
Graphical representation of viability of MCF-7 and U-118MG cells as a percentage after 24 h of incubation with Complexes **7** (**A**) and **5** (**B**).

**Figure 10 molecules-28-04609-f010:**
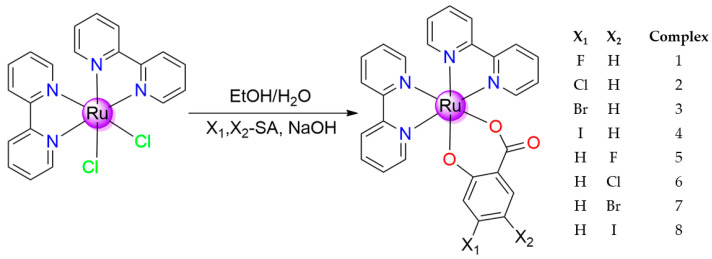
Schematic representation of the synthesis of **1**–**8**.

**Table 1 molecules-28-04609-t001:** Selected bands observed in FTIR spectra of prepared complexes.

Complex	*ν*(O-H)/cm^−1^	*ν*(C-H)_ar._/cm^−1^	*ν*(C=N)/cm^−1^	*ν_as_*(COO)/cm^−1^	*ν_s_*(COO)/cm^−1^	*δ*/cm^−1^	*ν*(C-O)/cm^−1^	*ν*(Ru-N)/cm^−1^
**1**	3234 (br, m)	3068 (w)	1609 (m)	1556 (s)	1416 (s)	140	1253 (s)	410 (w)
**2**	3356 (br, m)	3101, 3068 (w)	1583 (m)	1542 (s)	1415 (s)	127	1237 (s)	421 (w)
**3**	3354 (br, m)	3101, 3068 (w)	1578 (m)	1540 (s)	1416 (s)	124	1234 (s)	420 (w)
**4**	3294 (br, m)	3064 (w)	1571 (s)	1542 (s)	1413 (s)	129	1245 (s)	421 (w)
**5**	3383 (br, m)	3072 (w)	1603 (w)	1547 (s)	1412 (s)	135	1233 (s)	424 (w)
**6**	3384 (br, m)	3066 (w)	1589 (s)	1557 (s)	1402 (s)	155	1241 (m)	420 (w)
**7**	3320 (br, m)	3064 (w)	1584 (m)	1545 (s)	1400 (s)	145	1243 (s)	421 (w)
**8**	3359 (br, m)	3067 (w)	1571 (s)	1556 (s)	1417 (m)	139	1250 (s)	421 (w)

**Table 2 molecules-28-04609-t002:** Bands in the electron spectra of the prepared Complexes **1**–**8** and precursor complex [Ru(bipy)_2_Cl_2_] (**P1**) in nm.

Complex	ILCT I	ILCT II	^1^MLCT I	^1^MLCT II	MLCT I	MLCT II
**P1**	245	293	349	497	363	544 (583, 641)
**1**	245	295	363	524	369, 396	575 (688)
**2**	244	295	370	524	368, 394	571, 604 (708)
**3**	244	295	370	524	364, 393	575 (716)
**4**	264	295	367	524	360, 393	573 (711)
**5**	245	295	368	526	395	573 (725)
**6**	244	295	371	525	389, 503	602 (719)
**7**	244	295	370	524	362	601
**8**	242	295	366	525	391, 496	595 (742)

**Table 3 molecules-28-04609-t003:** Selected bond lengths (Å) in complex molecules for complexes **1**·3H_2_O·EtOH, **2**·2.6H_2_O·2EtOH, **3**·6H_2_O, and **4**·3H_2_O.

Compound	1	2	3	4
**Ru1-O1**	2.066 (5)	2.0587 (18)	2.070 (6)	2.081 (5)
**Ru1-O3**	2.067 (5)	2.0649 (18)	2.065 (6)	2.066 (5)
**Ru1-N1**	2.035 (6)	2.027 (2)	2.031 (7)	2.047 (6)
**Ru1-N2**	2.057 (7)	2.052 (2)	2.052 (8)	2.044 (6)
**Ru1-N3**	2.020 (7)	2.043 (2)	2.049 (8)	2.052 (5)
**Ru1-N4**	2.023 (6)	2.028 (2)	2.029 (7)	2.027 (6)
**Ru2-O4**	2.082 (5)	-	-	2.073 (5)
**Ru2-O6**	2.073 (5)	-	-	2.081 (5)
**Ru2-N5**	2.032 (6)	-	-	2.020 (6)
**Ru2-N6**	2.050 (7)	-	-	2.081 (5)
**Ru2-N7**	2.046 (7)	-	-	2.046 (5)
**Ru2-N8**	2.031 (6)	-	-	2.020 (6)

**Table 4 molecules-28-04609-t004:** Selected bond lengths (Å) for complexes **5**·1.55H_2_O, **7A**·1.75H_2_O, **7B**·H_2_O·EtOH and **8**·4H_2_O.

Compound	5	7A	7B	8
**Ru1-O1**	2.061 (7)	2.062 (4)	2.067 (3)	2.091 (4)
**Ru1-O3**	2.053 (6)	2.064 (4)	2.070 (3)	2.071 (4)
**Ru1-N1**	2.050 (8)	2.020 (4)	2.029 (3)	2.032 (5)
**Ru1-N2**	2.009 (8)	2.053 (4)	2.053 (3)	2.055 (5)
**Ru1-N3**	2.037 (8)	2.049 (4)	2.044 (3)	2.051 (5)
**Ru1-N4**	2.000 (8)	2.016 (4)	2.024 (3)	2.022 (5)

**Table 5 molecules-28-04609-t005:** Values of BSA fluorescence quenching rate constants (*k_q_*), dynamic quenching rate constants (*K_SV_*), binding constants (*K*), and number of binding sites per BSA (*n*) found for Complexes **1**–**8**.

Compound	Δ*I*/*I*_0_ (%)	*K_SV_* (M^−1^) × 10^5^	*k_q_* (M^−1^s^−1^) × 10^13^	*K* × 10^4^	*n*
**1**	69.7	1.21 (±0.05)	1.21 (±0.02)	7.32 (±0.34)	1.20
**2**	78.0	1.89 (±0.05)	1.89 (±0.02)	12.82 (±0.61)	1.12
**3**	75.1	1.61 (±0.03)	1.61 (±0.01)	14.80 (±0.41)	1.02
**4**	66.9	1.07 (±0.02)	1.07 (±0.01)	16.75 (±0.940)	0.86
**5**	73.0	1.45 (±0.06)	1.45 (±0.02)	7.23 (±0.250)	1.28
**6**	74.9	1.56 (±0.03)	1.56 (±0.01)	18.99 (±0.93)	0.93
**7**	79.1	1.99 (±0.03)	1.99 (±0.01)	22.42 (±0.27)	0.97
**8**	75.7	1.57 (±0.11)	1.57 (±0.05)	36.52 (±2.42)	0.87

**Table 6 molecules-28-04609-t006:** DNA binding constant and UV spectral features of **1**–**8** in the presence of DNA.

Compound	λ (nm)	Δ*A*/*A*_0_ (%)	Δ*λ* (nm)	*K_b_* (M^−1^)
**1**	245	40.5	+3	9.72 (±2.43) × 10^3^
**2**	244	20.0	+1	6.33 (±1.67) × 10^5^
**3**	244	18.8	+1	4.09 (±1.33) × 10^5^
**4**	264	20.3	−16	1.26 (±0.13) × 10^5^
**5**	245	11.9	+2	2.05 (±0.48) × 10^4^
**6**	244	20.6	+3	3.97 (±0.44) × 10^3^
**7**	244	14.8	+3	4.63 (±0.93) × 10^3^
**8**	242	24.2	+3	8.790 (±1.26) × 10^3^

**Table 7 molecules-28-04609-t007:** Values of relative changes in EB-DNA fluorescence *(*Δ*I*/*I*_0_), quenching rate constants (*k_q_*), dynamic quenching rate constants (*K_SV_*) for Complexes **1**–**8**.

Compound	Δ*I*/*I*_0_ (%)	*K_SV_* (M^−1^) × 10^4^	*k_q_* (M^−1^s^−1^) × 10^12^
**1**	85.2	3.15 (±0.23)	1.37 (±0.10)
**2**	94.7	10.70 (±0.98)	4.65 (±0.43)
**3**	91.1	6.41 (±0.55)	2.79 (±0.23)
**4**	83.5	4.76 (±0.41)	2.07 (±0.17)
**5**	92.0	7.10 (±0.58)	3.09 (±0.25)
**6**	79.4	6.68 (±0.83)	2.90 (±0.36)
**7**	94.4	9.39 (±1.16)	4.08 (±0.50)
**8**	86.2	5.22 (±0.51)	2.27 (±0.22)

**Table 8 molecules-28-04609-t008:** Table of IC(50) values of Complexes **1**–**8** against MCF-7 and U-118MG cell lines obtained during 72 h of incubation of cells with the complexes.

Compound	U-118MG			MCF-7		
	24 h	48 h	72 h	24 h	48 h	72 h
**1**	>10 × 10^−6^	>10 × 10^−6^	>10 × 10^−6^	5.76 × 10^−6^	4.75 × 10^−6^	3.30 × 10^−6^
**2**	>10 × 10^−6^	>10 × 10^−6^	>10 × 10^−6^	4.97 × 10^−6^	5.75 × 10^−6^	4.23 × 10^−6^
**3**	>10 × 10^−6^	>10 × 10^−6^	>10 × 10^−6^	>10 × 10^−6^	6.78 × 10^−6^	5.69 × 10^−6^
**4**	>10 × 10^−6^	>10 × 10^−6^	>10 × 10^−6^	4.92 × 10^−6^	6.18 × 10^−6^	6.65 × 10^−6^
**5**	3.56 × 10^−6^	4.72 × 10^−6^	3.49 × 10^−6^	<2 × 10^−6^	<2 × 10^−6^	<2 × 10^−6^
**6**	>10 × 10^−6^	>10 × 10^−6^	>10 × 10^−6^	>10 × 10^−6^	6.20 × 10^−6^	3.79 × 10^−6^
**7**	5.35 × 10^−6^	3.95 × 10^−6^	8.38 × 10^−6^	4.23 × 10^−6^	4.92 × 10^−6^	<2 × 10^−6^
**8**	4.08 × 10^−6^	2.65 × 10^−6^	>10 × 10^−6^	<2 × 10^−6^	<2 × 10^−6^	<2 × 10^−6^

**Table 9 molecules-28-04609-t009:** Crystallographic data for complexes in the adduct forms of crystals.

Compound	1·3H_2_O·EtOH	2·2.6H_2_O·2EtOH	3·6H_2_O	4·3H_2_O	5·1.55H_2_O·EtOH	7A·1.75H_2_O	7B·H_2_O·EtOH	8·4H_2_O
Chemical formula	C_27_H_21_FN_4_O_4_Ru	C_31_H_33_ClN_4_O_6_Ru	C_27_H_21_BrN_4_O_4_Ru	C_27_H_21.5_IN_4_O_4.25_Ru	C_29_H_31_FN_4_O_7_Ru	C_27_H_21_BrN_4_O_4_Ru	C_29_H_27_BrN_4_O_5_Ru	C_27_H_27_IN_4_O_7_Ru
M_r_	585.55	694.13	646.46	697.95	667.65	646.46	692.52	747.49
Crystal system	Monoclinic	Trigonal	Trigonal	Triclinic	Monoclinic	Triclinic	Triclinic	Monoclinic
Space group	*P*2_1_/c	*R* $\bar{3}$	*R* $\bar{3}$	$P\bar{1}$	*P*2_1_/c	$P\bar{1}$	$P\bar{1}$	*P*2_1_/c
T/K	100	100	100	100	100	100	100	100
*a*/Å	15.5799 (6)	25.4969 (2)	25.7119 (7)	11.3038 (2)	10.2179 (2)	9.4768 (6)	9.79637 (20)	9.9270 (5)
*b*/Å	24.1583 (9)	25.4969 (2)	25.7119 (7)	16.1443 (4)	26.3315 (6)	9.9798 (6)	10.27595 (16)	27.7411 (12)
*c*/Å	14.9993 (5)	24.4547 (2)	24.1522 (6)	16.2832 (3)	10.8839 (2)	14.9267 (5)	14.58477 (17)	11.0554 (6)
*α*/°	90	90	90	79.887 (2)	90	99.343 (4)	73.1374 (12)	90
*β*/°	101.057 (3)	90	90	71.323 (2)	106.823 (2)	90.086 (4)	79.6472 (15)	112.761 (6)
*γ*/°	90	120	120	74.945 (2)	90	112.648 (6)	87.7349 (15)	90
*V*/Å^3^	5540.7 (4)	13767.9 (2)	13827.9 (8)	2704.55 (11)	2803.02 (10)	1282.39 (13)	1382.08 (4)	2807.4 (3)
*Z*	8	18	18	2	4	2	2	4
*μ*/mm^−1^	4.959	5.369	5.942	13.995	5.058	7.119	6.676	13.602
Crystal size/mm	0.12 × 0.15 × 0.24	0.13 × 0.22 × 0.22	0.15 × 0.18 × 0.23	0.05 × 0.08 × 0.12	0.01 × 0.25 × 0.38	0.18 × 0.21 × 0.24	0.24 × 0.33 × 0.33	0.13 × 0.14 × 0.18
*ρ_calc_*/g·cm^−3^	1.404	1.507	1.397	1.714	1.582	1.674	1.664	1.769
*S*	1.046	1.033	1.068	1.047	1.185	1.060	1.070	1.023
*R_1_* [*I* > 2*σ*(*I*)]	0.0763	0.0391	0.0549	0.0648	0.0869	0.0478	0.0437	0.0511
*wR_2_* [All data]	0.2088	0.1076	0.1473	0.1800	0.2456	0.1296	0.1172	0.1358
Largest diff. peak/hole e/Å^−3^	1.82/−1.44	0.70/−0.83	2.38/−1.60	2.98/−2.97	2.93/−1.62	1.27/−1.87	1.20/−1.38	1.04/−1.14
CCDC	2259382	2259383	2259384	2259385	2259386	2259387	2259388	2259389

## Data Availability

Not applicable.

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
