# Peer review of "Bipyridine Ruthenium(II) Complexes with Halogen-Substituted Salicylates: Synthesis, Crystal Structure, and Biological Activity"

_molecules, 2023, doi:10.3390/molecules28124609_

Round 1

Reviewer 1 Report

The manuscript from the group of Moncol reports the preparation of a family of ruthenium bipyridine complexes containing halide-substituted salicylate ligands. The compounds have been comprehensively characterized by a variety of different techniques. The complexes were found to show solvatochromic behaviour, which was studied in a range of solvents by UV-Vis-spectroscopy. The molecular structures were also determined by X-ray diffraction. In addition, interaction of the complexes with BSA and DNA was examined by fluorescence spectroscopy. Finally, in vitro cytotoxicity was studied in two cancer cell lines.

The manuscript is well presented and the work is competently carried out. The manuscript should be accepted after some very minor revisions:

In the crystal structure 4 an incorrect value of Z is given. This should be corrected.

In order to get a better idea of cytotoxicity it would be nice to see data for other benchmark anticancer drugs for comparison.

Author Response

  1. In the crystal structure 4 an incorrect value of Z is given. This should be corrected.

Answer: Thank you very much for reviewing the article. The value of parameter Z has been corrected in the manuscript.

  1. In order to get a better idea of cytotoxicity it would be nice to see data for other benchmark anticancer drugs for comparison.

Answer: The study of the properties of the complexes from the manuscript is still ongoing. The study of spectro-electrochemical properties and a more detailed study of the biological properties is currently in progress. For the MTT assay, we also plan to measure clinically used therapeutics - we currently have the capability to measure 5-fluorouracil and we plan to use cisplatin as well.

Reviewer 2 Report

The manuscript submitted by M. Shoeller et al. is devoted to a series of new octahedral ruthenium(II) complexes combining bipyridine and halogen-substituted salicylate ligands. The latter were chosen owing to their relevance to anti-inflammatory drugs, and this really seems like an interesting idea. The synthesis and characterization of the resulting complexes were supplemented by bioactivity studies which eventually revealed remarkable antiproliferative activity of at least some of them. The manuscript can be accepted for publication after some minor changes according to the following comments.

1. Section 2.1 should be supplemented by the scheme of synthesis of the mentioned complexes with their general structural formula (indicating the substituents in the salicylate ligand).

2. The addition and discussion of the 13C NMR spectroscopic data (e.g., for complex 6) seem to be reasonable.

3. The manuscript conclusions lack the information about future investigations which the authors are planning to perform (directions of further ligand design, mechanistic studies, etc.). What is the potential of this type of Ru(II) complexes?

4. Some unfortunate phrases, errors, typos:

P. 1, lines 22-23: "The structure of the complexes was determined by X-ray structural analysis and NMR spectroscopy. All complexes were characterized by spectral methods." (the second sentence partially duplicates information of the first sentence);

P. 2, line 63: can be describe --> can be described;

P. 4, line 155: "The 1H NMR" ("1" should be moved to superscript);

P. 5, line 201: hypochromic effect --> hypsochromic effect;

P. 7, line 211: coordinate bidentate --> coordinate bidentately;

P. 9, line 262: hirshfeld --> Hirshfeld.

Minor editing of English language is required.

Author Response

  1. Section 2.1 should be supplemented by the scheme of synthesis of the mentioned complexes with their general structural formula (indicating the substituents in the salicylate ligand).

Answer: Thank you very much for reviewing the article. We have added a scheme for the synthesis of the complexes to the manuscript. We are grateful for this suggestion. We believe that the addition of the scheme will make the synthesis of the complexes clearer for the reader.

  1. The addition and discussion of the 13C NMR spectroscopic data (g., for complex 6) seem to be reasonable.

Answer: The complexes were not sufficiently soluble in DMSO to measure 13C NMR spectra of adequate quality to be published. Spectra were measured for several complexes, but a few peaks were unrecognisable due to high background signal.

  1. The manuscript conclusions lack the information about future investigations which the authors are planning to perform (directions of further ligand design, mechanistic studies, ). What is the potential of this type of Ru(II) complexes?

Answer: The study of the properties of the complexes from the manuscript is still ongoing. The study of spectro-electrochemical properties and a more detailed study of the biological properties is currently in progress. Thank you for the suggestion, we have added this information to the conclusion.

  1. Some unfortunate phrases, errors, typos:
  2. 1, lines 22-23: "The structure of the complexes was determined by X-ray structural analysis and NMR spectroscopy. All complexes were characterized by spectral methods." (the second sentence partially duplicates information of the first sentence);
  3. 2, line 63: can be describe --> can be described;
  4. 4, line 155: "The 1H NMR" ("1" should be moved to superscript);
  5. 5, line 201: hypochromic effect --> hypsochromic effect;
  6. 7, line 211: coordinate bidentate --> coordinate bidentately;
  7. 9, line 262: hirshfeld --> Hirshfeld.

Answer: Thank you for noticing, we have corrected all the mentioned errors and typos in the text of the manuscript.

Reviewer 3 Report

The paper of Ján Moncol and co-authors is an interesting fundamental work on synthesis 8 new Ruthenium complexes with diimine (2,2’-bipyridine) and modified salicylates. Authors succeeded growing crystals amenable for SC X-RAY analysis for 7 compounds, determined their crystal structure. All complexes were studied using NMR and IR spectroscopy, mass spectrometry. Their structure is not in doubt. The article is written very well, neatly, I have no major remarks. However, there are a few questions and suggestions that, in my humble opinion, can improve the text of the manuscript.

From the introduction of the article, it is not clear why the authors paid attention to diimine ligands and their complexes. There are a few lines talking about diimines in the introduction. At the same time, an easy search indicated several papers on diimine complexes and their anticancer activity. It may be worth paying attention to these works and supplementing the introduction in order to make it clear to readers why diimines were chosen as ligands.

10.1039/C8NJ04195D - α-Diimine homologues of cisplatin: synthesis, speciation in DMSO/water and cytotoxicity

10.3390/molecules27238565 - α-Diimine Cisplatin Derivatives: Synthesis, Structure, Cyclic Voltammetry and Cytotoxicity

10.1021/acs.inorgchem.1c03314 - Heteroleptic Pd(II) and Pt(II) Complexes with Redox-Active Ligands: Synthesis, Structure, and Multimodal Anticancer Mechanism

I hope, it will be useful for the authors of the manuscript to know that diimines are so-called redox-active ligands (that is, they can reversibly accept electrons) 10.1016/j.ccr.2010.01.009 «Non-innocent ligands in bioinorganic chemistry—An overview». And since Ruthenium ions are also redox active, it would be very interesting to observe the electrochemical measurements of the obtained complexes in the next papers of the authors. It is possible that the resulting radicals or the desired redox potential will play a key role in the biological behavior of Ruthenium complexes with diimines for example 10.3390/molecules27238565 - α-Diimine Cisplatin Derivatives: Synthesis, Structure, Cyclic Voltammetry and Cytotoxicity or 10.1016/j.ccr.2022.214875, 10.1002/ejic.201600908. 

Lines 98-104: Citing the work as an example, nothing is said about diimine ligands. As if they are only “innocent” ligands.

Line 123: “All complexes have been purified by column chromatography.” - What did the authors purify the reaction products from? isomers? are they stereoisomers and can they be separated on a column? Or impurities? If impurities, what is their composition according to the authors?

In section 2.1.1. When studying the IR spectra of compounds, much attention is paid to the vibrations of chemical bonds in salicylic groups. The presence of diimine ligands is indirectly confirmed by vibrations of ruthenium-nitrogen bonds (+ C-H). At the same time, diimine ligands have characteristic C=N bonds, which can be found in the spectra. And it can also pay attention that most likely these bands will be partially shifted towards lower frequencies during coordination. It is also worth paying attention to the vibrations of bonds in the aromatic rings of bipyridine.

What role do diimine ligands and salicylic residues play in binding to DNA? For example, platinum-chlorine bonds play an important role in cisplatin However, in DMSO, cisplatin is unstable: chloride ions are exchanged with the solvent; therefore, it is still important to synthesize new complexes with diimine ligands that affect the dissociation energy of the platinum-chlorine bond.

Line 450: “The solutions thus prepared were mixed and allowed to reflux for three days.” - I wonder why three days. Why is the reaction taking so long, professional interest.

For obtained complexes, the electrical conductivity of solutions was studied. What was it for? what can be said, what conclusion can be drawn based on the electrical conductivity?

I hope that the references I have provided will help the authors improve the introduction or the biological part of the manuscript, as well as correct further work with redox-active diimine ligands.

The presented manuscript is a very interesting work and I certainly recommend accepting it for publication in the journal Molecules. This work will definitely attract a lot of attention from researchers and I hope increase citations for the Molecules.

Author Response

  1. From the introduction of the article, it is not clear why the authors paid attention to diimine ligands and their complexes. There are a few lines talking about diimines in the introduction. At the same time, an easy search indicated several papers on diimine complexes and their anticancer activity. It may be worth paying attention to these works and supplementing the introduction in order to make it clear to readers why diimines were chosen as ligands.

Answers to 1 +2 +3: Thank you very much for reviewing the article. In this paper, the ligand 2,2-bipyridine was chosen primarily because of its planar aromatic structure, which allows it to interact with the DNA intercalation mechanism. In points 1 and 2 you describe the electrochemical properties of the non-innocent ligands. However, the object of this article is not electrochemistry. This study is in progress on these complexes, and we plan to publish this study later. Currently, the complexes are studied by spectro-electrochemical methods - Voltammetry + UV-Vis-NIR and EPR. We did not discuss this issue in detail in the introduction, as we do not yet report these results in this paper. However, we are grateful for the advice. We believe that if our next steps in the study of these substances were listed at the end of the manuscript, it would be clearer for the reader. We have therefore added this to the conclusion.

  1. I hope, it will be useful for the authors of the manuscript to know that diimines are so-called redox-active ligands (that is, they can reversibly accept electrons) 10.1016/j.ccr.2010.01.009 «Non-innocent ligands in bioinorganic chemistry—An overview». And since Ruthenium ions are also redox active, it would be very interesting to observe the electrochemical measurements of the obtained complexes in the next papers of the authors. It is possible that the resulting radicals or the desired redox potential will play a key role in the biological behavior of Ruthenium complexes with diimines for example 10.3390/molecules27238565 - α-Diimine Cisplatin Derivatives: Synthesis, Structure, Cyclic Voltammetry and Cytotoxicity or 10.1016/j.ccr.2022.214875, 10.1002/ejic.201600908.

Answers to 1 +2 +3: Thank you very much for reviewing the article. In this paper, the ligand 2,2-bipyridine was chosen primarily because of its planar aromatic structure, which allows it to interact with the DNA intercalation mechanism. In points 1 and 2 you describe the electrochemical properties of the non-innocent ligands. However, the object of this article is not electrochemistry. This study is in progress on these complexes, and we plan to publish this study later. Currently, the complexes are studied by spectro-electrochemical methods - Voltammetry + UV-Vis-NIR and EPR. We did not discuss this issue in detail in the introduction, as we do not yet report these results in this paper. However, we are grateful for the advice. We believe that if our next steps in the study of these substances were listed at the end of the manuscript, it would be clearer for the reader. We have therefore added this to the conclusion.

  1. Lines 98-104: Citing the work as an example, nothing is said about diimine ligands. As if they are only “innocent” ligands.

Answers to 1 +2 +3: Thank you very much for reviewing the article. In this paper, the ligand 2,2-bipyridine was chosen primarily because of its planar aromatic structure, which allows it to interact with the DNA intercalation mechanism. In points 1 and 2 you describe the electrochemical properties of the non-innocent ligands. However, the object of this article is not electrochemistry. This study is in progress on these complexes, and we plan to publish this study later. Currently, the complexes are studied by spectro-electrochemical methods - Voltammetry + UV-Vis-NIR and EPR. We did not discuss this issue in detail in the introduction, as we do not yet report these results in this paper. However, we are grateful for the advice. We believe that if our next steps in the study of these substances were listed at the end of the manuscript, it would be clearer for the reader. We have therefore added this to the conclusion.

  1. Line 123: “All complexes have been purified by column chromatography.” - What did the authors purify the reaction products from? isomers? are they stereoisomers and can they be separated on a column? Or impurities? If impurities, what is their composition according to the authors?

Answer and comments: During chromatographic purification of the complexes, we also tried to isolate the "impurities". However, our efforts to identify them were unsuccessful. The substances were strongly coloured, i.e. we believe that they are by-products of synthesis (2xsalicylato+1xbipy complex, multinuclear complexes, 3xbipy complex, unreacted starting reagents). With such intense staining, we thought the isolation would be successful, but after evaporation of the solvent, there was not a significant amount of substance left in the flask. We also attach photo-documentation, where the "impurities" were not enough quantity to be identified by the available experimental methods.

  1. In section 2.1.1. When studying the IR spectra of compounds, much attention is paid to the vibrations of chemical bonds in salicylic groups. The presence of diimine ligands is indirectly confirmed by vibrations of ruthenium-nitrogen bonds (+ C-H). At the same time, diimine ligands have characteristic C=N bonds, which can be found in the spectra. And it can also pay attention that most likely these bands will be partially shifted towards lower frequencies during coordination. It is also worth paying attention to the vibrations of bonds in the aromatic rings of bipyridine.

Answer: Thank you for the advice, we have added the C=N vibration wavenumbers to the table.

  1. What role do diimine ligands and salicylic residues play in binding to DNA? For example, platinum-chlorine bonds play an important role in cisplatin However, in DMSO, cisplatin is unstable: chloride ions are exchanged with the solvent; therefore, it is still important to synthesize new complexes with diimine ligands that affect the dissociation energy of the platinum-chlorine bond.

Answer: All results suggest that there is a non-covalent interaction mediated by bipy ligands - intercalation.

  1. Line 450: “The solutions thus prepared were mixed and allowed to reflux for three days.” - I wonder why three days. Why is the reaction taking so long, professional interest.

Answer and comments: Since ruthenium complexes are kinetically inert, syntheses under "normal" conditions take longer. Various advanced methods, for example based on reactions in microwave reactors, are often used. In our case, we used a reaction medium - mixed solvent EtOH+water, which did not allow heating at higher temperatures. The reactions were monitored by TLC, the aim being to be sure that an equilibrium state was reached in the reaction system - maximum confidence that the product yield would no longer be higher. This is why such a long reaction time was chosen - to have maximum certainty of maximising the yield - it is possible that even 24 hours would have been sufficient.

  1. For obtained complexes, the electrical conductivity of solutions was studied. What was it for? what can be said, what conclusion can be drawn based on the electrical conductivity?

The electrical conductivity was measured only to confirm that the complex retains its "integrity" even in DMSO solution. that there is no rapid dissociation to form charged particles.

I hope that the references I have provided will help the authors improve the introduction or the biological part of the manuscript, as well as correct further work with redox-active diimine ligands.

Thank you once again for the stimulating articles that were used in the following article dealing with electrochemistry and spectro-electrochemistry.
